# *MTOR* Variation Related to Heat Resistance of Chinese Cattle

**DOI:** 10.3390/ani9110915

**Published:** 2019-11-04

**Authors:** Qingqing Ning, Kaixing Qu, Quratulain Hanif, Yutang Jia, Haijian Cheng, Jicai Zhang, Ningbo Chen, Hong Chen, Bizhi Huang, Chuzhao Lei

**Affiliations:** 1Key Laboratory of Animal Genetics, Breeding and Reproduction of Shaanxi Province, College of Animal Science and Technology, Northwest A&F University, Yangling 712100, China; 18392375865@139.com (Q.N.); ningboch@126.com (N.C.); chenhong1212@263.net (H.C.); 2Yunnan Academy of Grassland and Animal Science, Kunming 650212, China; kaixqu@163.com (K.Q.); ynzjc@126.com (J.Z.); 3National Institute for Biotechnology and Genetic Engineering, Pakistan Institute of Engineering and Applied Sciences, Faisalabad 577, Pakistan; micro32uvas@gmail.com; 4Institute of Animal Science and Veterinary Medicine, Anhui Academy of Agriculture Science, Hefei 230001, China; yutang2018@163.com; 5Institute of Animal Science and Veterinary Medicine, Shandong Academy of Agricultural Sciences, Jinan 250100, China; 98061107@163.com

**Keywords:** Chinese cattle, *MTOR* gene, variation, heat tolerance, association

## Abstract

**Simple Summary:**

Due to unique geographical distribution and appearance characteristics, China cattle has been divided into three groups: northern cattle (dominated by *Bos taurus* in northern China), central cattle (admixture of *Bos taurus* and *Bos indicus* in the middle region) and southern cattle (dominated by *Bos indicus* in southern China). With this rule, it was believed that southern in cattle are more heat resistant than northern cattle. Previous studies showed that the mechanistic target of the rapamycin (*MTOR)* (NC_037343.1:c.2062G>C) gene could be associated with heat resistance. This study used PCR and sequencing to type this locus in 1030 individuals of 37 cattle breeds and proved the mutation of this locus could be related to heat tolerance in Chinese cattle.

**Abstract:**

With the inexorable rise of global temperature, heat stress deserves more and more attention in livestock agriculture. Previous studies have shown that the mechanistic target of rapamycin (*MTOR*) (NC_037343.1:c.2062G>C) gene contributes to the repair of DNA damage repair and is associated with the adaptation of camels in dry and hot environments. However, it is unknown whether this mutation is related to the heat tolerance of Chinese cattle. In this study, PCR and sequencing were used to type the mutation locus in 1030 individuals of 37 cattle breeds. The analysis results showed that the frequency of G allele of the locus gradually diminished from the northern group to the southern group of native Chinese cattle, whereas the frequency of the C allele showed an opposite pattern, displaying a significant geographical difference across native Chinese cattle breeds. Additionally, an analysis of the locus in Chinese indigenous cattle revealed that this SNP was significantly associated with mean annual temperature (T), relative humidity (RH) and temperature humidity index (THI) (*p* < 0.01), suggesting that cattle with C allele was distributed in regions with higher T, RH and THI. In conclusion, this study proved that the mutation of *MTOR* gene in Chinese cattle could be associated with the heat tolerance.

## 1. Introduction

Environments of high temperatures and humidity are detrimental to the productivity of commercial animal agriculture [1,2]. Therefore, in this case, heat stress in cattle negatively impacts on animal production, impairs normal bodily function, jeopardizes animal welfare [3,4,5,6,7] and also causes massive livestock economy losses [5]. However, the detrimental effects of HS (heat stress) will likely become more serious if the earth’s climate continues to warm as predicted [8].

The indiscriminate emission of greenhouse gases not only causes global warming but also destroys the ozone layer and leads to increasing ultraviolet radiation. Based on previous research, mammalian cells exposed to environmental stresses require an efficient DNA repair mechanism to maintain genomic instability [9]. Coincidently, the mechanistic target of rapamycin (*MTOR*) has roles related to DNA damage repair, and its mechanism has been confirmed by many studies. Dominick et al. (2017) proved that a novel link between DNA repair and mTOR signaling via post-transcriptional regulation and mRNA level involving specific alteration in the CCR4-NOT complex, which means if CCR4-NOT complex as downstream of *MTOR* gene in regulating NDRG1 (N-myc downstream-regulated gene 1) and MGMT (O-6-methylguanine-DNA methyltransferase) expression is reduced or inactivated, it will lead to improve DNA repair capacity [10,11]. Reiter et al. (2004) reported that mTOR can also regulate stress resistance by preferential translation of certain mRNAs [12]. Furthermore, this gene has been identified as the rapidly evolving genes in camels, which results in insulin resistance via serine phosphorylation of insulin receptor substrates proteins to store energy and hold a high-level blood glucose. Further, this is conducive to survive in hot and arid deserts or semi-deserts [13,14].

In line with the above studies regarding heat resistance, the related gene located in 16 chromosome (NC_037343.1) about *Bos taurus* was found by the National Center for Biotechnology Information (NCBI). In the meantime, this study went on to search for a SNP (mutation frequency over 0.2, rs445599276) located in ~1.1kb downstream of the *MTOR* gene from the Bovine Genome Variation Database and Selective Signatures (BGVD) [15], and more importantly whose allele frequency distribution of Chinese cattle breeds were significant geographical distribution rules (such as Dianzhong (6):0.750 in southern China but Chaidamu (5):0.000 and Tibetan (9):0.000 in northern China). Taken together, it has a selective advantage to speculate that the *MTOR* mutation might be linked to the thermo-tolerance.

China is one of the countries with the most abundant livestock and poultry genetic resources in the world, containing more than 53 cattle breeds [16]. China’s geographical address is located across the cold temperate, temperate and subtropical zone, which contributes to different temperature and humidity from north to south. Therefore, due to its unique geographical distribution and appearance characteristics, it is divided into three groups: northern cattle (dominated by *Bos taurus* in northern China), central cattle (admixture of *Bos taurus* and *Bos indicus* in the middle region) and southern cattle (dominated by *Bos indicus* in southern China) [17]. Additionally, as a new generation of molecular genetic markers, SNPs are of great value for the animals’ genetic diversity and structure, origin, phenotypic correlation analysis and so forth [18].

From the above, Chinese cattle breeds are very suitable for the detection of SNP in the bovine *MTOR* gene and for testing the relationship between *MTOR* variants and the mean annual temperature (T), relative humidity (RH) and the temperature humidity index (THI) of a sampling site. It is expected that results of this study will provide basic data for marker-assisted selection related to the heat tolerance.

## 2. Materials and Methods

### 2.1. Ethics Statement

The protocols used in this study for the animals were recognized by the Faculty Animal Policy and Welfare Committee of the Northwest A&F University (FAPWC-NWAFU, protocol number, NWAFAC1008).

### 2.2. Animal Samples Information, DNA Extraction, and Data Collection

The genomic DNA of 1030 individuals representing 35 Chinese native cattle breeds as well as Angus and Burmese cattle as controls (Appendix A) was extracted from ear tissue samples using standard phenol–chloroform method [19]. The controls were identified relatively as pure *Bos taurus* and *Bos indicus* according to previous studies which proved: *Bos indicus* possessed the best heat resistance [20], whereas *Bos taurus* did not have thermal mutations at all [21,22]. Then, the extracted genomic DNA concentration was determined by spectrophotometry, correspondingly diluted to 10 ng/µL and finally, stored at −80 °C until use.

Three environmental parameters (T, RH, THI) at the sampling sites of 35 native cattle breeds in the past 30 years (from 1971 to 2000) were collected by the Chinese Central Meteorological Agency (http://data.cma.cn/) (Appendix A).

### 2.3. Primers Information, PCR Amplification and PCR Product Processing

The primers of *MTOR* gene locus were designed through NCBI and used to type this SNP. The primer sequences, fragment sizes and annealing temperatures are shown in Appendix A. Each 25 μL PCR amplification mixture contained 20 ng of genomic DNA, 20 pmol/μL of each primer, 0.2 mmol of dNTPs, 1× PCR buffer (including 2.5 mmol of Mg^2+^) and 1.0 U of rTaq DNA polymerase (Takara, Dalian, China). A thermocycling protocol was 5 min at 95 °C and 35 cycles of 94 °C for 30 s, annealing at 54 °C for 30 s and at 72 °C for 30 s (Appendix A), with a final extension at 72 °C for 10 min. The PCR products of all 1030 samples were detected by electrophoresis on a 2.0% agarose gel stained with ethidium bromide. Then, PCR products were directly sequenced with an ABI PRIZM 377 DNA sequencer (PerkinElmer). All tests were conducted at Shanghai Sangon Biotech Company, Shanghai, China. The sequencing results were read by Chromas 2.6.5 Software (Technelysium Pty Ltd., South Brisbane, Queensland, Australia).

### 2.4. Statistic Analysis of MTOR Gene Polymorphism

According to the Hardy-Weinberg equilibrium (HWE), the genotypic and allele frequencies were calculated directly based on the genotypes observed in the analyzed breeds. Then, the allele frequency of each breed would be depicted on the map of China by a pie chart in order to observe its distribution in Chinese cattle (Figure 1). The PowerMarker v3.25 software (North Carolina State University, Raleigh, NC, USA) was used to calculate and analyze the allele frequencies, alleles per locus (Ne), observed heterozygosity (Ho) and Nei’s expected heterozygosity (He) for the *MTOR* locus [23]. Then, the Hardy-Weinberg balance was tested by the program POPGENE Version 1.31 (University of Calgary, Calgary, AB, Canada) [24,25]. The polymorphism was measured by the polymorphism information content (PIC) value for the *MTOR* locus [26].

The Hardy-Weinberg equilibrium (HWE) formula as follows:p + q = 1 and p² + 2 × p × q + q² = 1(1)

The G allele frequency of *MTOR* gene is p. The C allele frequency of *MTOR* gene is q.
(2)$$H_{0}\;=\sum_{i\;=\;1}^{n}P_{i}^{2}H_{e}=1-\sum_{i\;=\;1}^{n}P_{i}^{2}\mathrm{Ne}=1/\sum_{i\;=\;1}^{n}P_{i}^{2}$$
(3)$$\mathrm{PIC}\;=\;1-\sum_{i\;=\;1}^{m}P_{i}^{2}-\sum_{i\;=\;1}^{m\;-\;1}\sum_{j\;=\;i\;+\;1}^{m}2P_{i}^{2}P_{j}^{2}$$

In our study, THI was calculated based on the formula used by the National Oceanic and Atmospheric Administration (1976) [27]:THI = (1.8T + 32) − (0.55 − 0.0055RH) × (1.8T − 26)(4)
where T is temperature in degrees Celsius and RH is relative humidity as a percentage.

The least square means (LSM) of the distribution of the three environmental parameters (T, RH, THI) of the 35 Chinese cattle breeds were calculated using a general linear model using a IBM SPSS Modeler 23.0 Software (SPSS, Inc., Armonk, NY, USA):Y_i_ = μ + G_i_ + B_i_ + e_i_(5)
where Y_i_ is the value of T, RH and THI between 1971 and 2000, μ is the overall mean value, G_i_ is the fixed genotype effect, B_i_ is the fixed effect of breeds and e_i_ is a random residual effect. The differences were considered significant at *p* < 0.05.

## 3. Results

### 3.1. Diversity Analysis

The H_0_, H_e_, N_e_ and PIC of 37 breeds cattle in *MTOR* (NC_037343.1:c.2062G>C) locus were calculated by the relevant formulas (Table 1). From our research, H_0_ was detected from 0.5050 to 1.000 and H_e_ was from 0.0783 to 0.4950, which meant that they were not subject to high intensity selection and had rich genetic diversity [28]. According to the site polymorphism criteria determined by Botstein, there were twenty-nine loci which were reasonable polymorphism (0.5 > PIC > 0.25), eight were only slight polymorphism (PIC < 0.25) and no one had high polymorphism (PIC > 0.5). Furthermore, both of the moderate and low polymorphisms were distributed in the northern, central and southern populations. From the results of the test of the Hardy-Weinberg equilibrium for genetic diversity, the genotypic frequencies of GG, GC and CC were respectively 51.34%, 37.53% and 11.13%, and the results of the χ^2^ test indicated that there were only three breeds (Yanbian, Tibetan and Angus) having significant differences (*p* < 0.05), indicating that other 34 breeds all reached the genetic balance (*p* > 0.05).

### 3.2. Statistical Analysis of Genotypic and Allele Frequencies

The Angus and Burmese cattle were used as the controls to determine the allele frequencies for the *B. taurus* and *B. indicus* cattle populations. The allele frequency was obtained by counting genotype frequencies (Table 2 and Figure 1). The results showed that the C allele frequency of *MTOR* gene reached 71.67% in the Burma population, whereas all Angus cattle carried the G allele (Table 2 and Figure 1).

A base transversion (NC_037343.1:c.2062G>C) mutation was detected in the *MTOR* gene. This study then explored the allelic and genotypic frequencies of the locus in Chinese cattle, as shown in Table 2. At the *MTOR*: c.2062G>C locus, three genotypes (GG, GC and CC) were detected and the mean allelic frequencies for the G allele and C allele in Chinese cattle were 70.10% and 29.90%, respectively. In terms of the detailed geographic distribution, the frequencies of the C allele for the northern, central and southern groups were 0.1134, 0.2500 and 0.3551 respectively, and were gradually diminishing in the Chinese indigenous cattle from south to north.

### 3.3. Correlation Analysis of MTOR Gene Polymorphism

The results of the association analysis by one-way ANOVA between the genotypes and the three environmental parameters (T, RH and THI) for 35 breeds in 970 Chinese indigenous cattle were shown in Appendix A. At the *MTOR*: c.2062G>C locus, there was a significant association between the allele C and T, RH and THI. The individuals with CC or GC genotypes were found in areas with significantly higher T, RH and THI values compared with those having the GG genotype (*p* < 0.01) (Figure 1), suggesting that the allele C might be associated with heat tolerance in Chinese indigenous cattle. The test of effects of the three study parameters on *MTOR* genotype showed that the mean annual temperature had the strongest correlation with the genotypes (Table 3 and Appendix A).

## 4. Discussion

In recent years, the frequency of thermal diseases in animals has risen with the increase of the global temperature. Previous studies reported that the heat illness, a continuum of disorders caused by hyperthermia, includes heat cramps, heat exhaustion, heat injury and heat stroke [29,30,31], and the systemic inflammatory response syndrome (SIRS) is considered to be the primary cause of organ dysfunction related to heat stroke [32,33]. Therefore, the selection of heat-resistant traits is necessary in the commercial cattle industry. Additionally, previous study results also showed that the rapidly evolving *MTOR* gene for camels plays an important role in adapting the dryness-heat environment [13]. Furthermore, the *MTOR* gene had been proved to repair DNA damage via post-transcriptional regulation and the mRNA level involving specific alteration in the CCR4-NOT complex (located in the downstream of the mTORC1 pathway), which regulates N-myc downstream-regulated gene 1 and O-6-methylguanine-DNA methyltransferase to effect the DNA damage repair [10,11,12]. The *MTOR* gene was just exact for the cattle to maintain genomic instability when exposed to excessive ambient temperature. Therefore, we speculated that *MTOR* was an important candidate gene responsible for heat tolerance in Chinese cattle.

Furthermore, the distribution of allelic frequencies in the control groups showed that the allele G of the *MTOR* gene is dominant in *B.taurus* cattle (Angus), whereas the C allele is dominant in *B. indicus* cattle (Burmese zebu). A similar pattern was observed in the Chinese native cattle population. The results also showed that the frequencies of G allele of the *MTOR* gene diminished gradually in native Chinese cattle from the northern group to the southern group, whereas the frequencies of the C allele showed an opposite pattern (Figure 1), indicating a significant geographical difference across native Chinese cattle breeds and consistent with the distribution of indicine and taurine cattle in China. For the southern group, the highest frequencies of the C allele of the *MTOR* gene were found in cattle in southwestern China, which has the highest temperatures compared to other regions, followed by cattle in southeastern China. Correspondingly, this study also found that cattle breeds with particularly high C allele frequency had higher mean annual temperatures in the southern region. The results were consistent with previous studies that revealed the geographical environment and origin of Chinese indigenous cattle [21,22,34,35].

## 5. Conclusions

In conclusion, the results indicated that the variant of the *MTOR* gene was associated with heat tolerance in Chinese cattle and could be used in the marker-assisted selection program to improve the heat tolerance trait of Chinese cattle.

## Figures and Tables

**Figure 1 animals-09-00915-f001:**
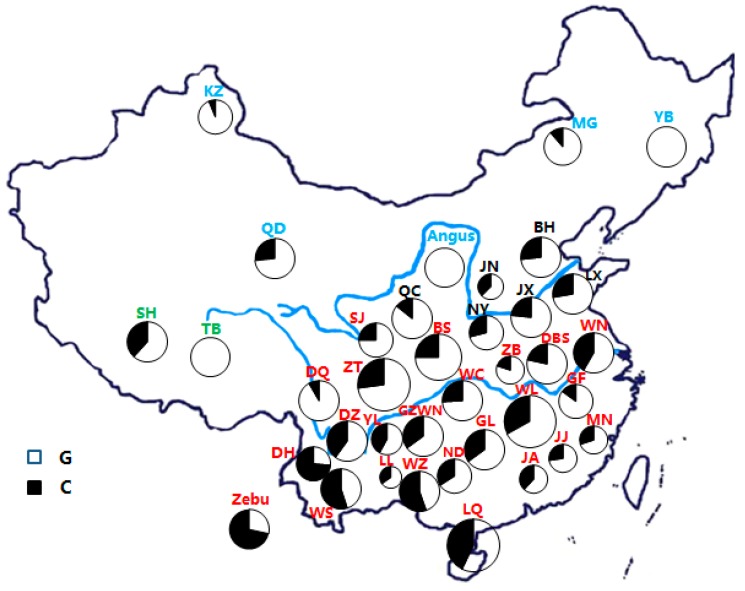
(a) Geographical distribution of two variants among 35 Chinese breeds as well as Augus and Burma populations. (b) Distribution of G and C alleles of the NC_037343.1:c.2062G>C loci of the *MTOR* gene. (c) YB, Yanbian; MG, Mongolian; KZ, Kazakh; QD, Chaidamu; SH, Shigatse Humped; TB Tibetan; BH, Bohai Black; JN, Jinnan; JX, Jiaxian Red; QC, Qinchuan; LX, Luxi; NY, Nanyang; WN, Wannan; DBS, Dabieshan; ZB, Zaobei; BS, Bashan; WC, Wuchuan; SJ, Sanjiang; ZT, Zhaotong; DQ, Diqing; DZ, Dianzhong; YL, Yunling; GZWN, Weining; GL, Guanling; GF, Guangfeng; MN, Minnan; JJ Jinjiang; JA, Ji’an; LQ, Leiqiong; WZ, Weizhou; ND, Nandan; LL, Longlin; DH, Dehong; WS, Wenshan; WL, Wuling.

**Table 1 animals-09-00915-t001:** Genetic indices *p*-value, Ho, He, Ne and polymorphism information content (PIC) of the *MTOR* gene across 35 Chinese cattle breeds as well as Angus and Burma.

Breeds	*p*-Value	Ho	He	Ne	PIC
Kazakh (KZ)	0.0625	0.8830	0.1170	1.1330	0.1103
Yanbian (YB)	0.0000	1.0000	0.0000	1.0000	0.0000
Chaidamu (QD)	0.2667	0.6089	0.3911	1.6423	0.3146
Mongolian (MG)	0.1143	0.7976	0.2024	1.2538	0.1820
Jiaxian Red (JX)	0.2333	0.6422	0.3578	1.5571	0.2938
Nanyang (NY)	0.2895	0.5886	0.4114	1.6988	0.3267
Luxi (LX)	0.2759	0.6005	0.3995	1.6653	0.3197
Bohai Black (BH)	0.2667	0.6089	0.3911	1.6423	0.3146
Jinnan (JN)	0.3750	0.5313	0.4688	1.8824	0.3589
Qinchuan (QC)	0.1429	0.7551	0.2449	1.3243	0.2149
Ji’an (JA)	0.3824	0.5277	0.4723	1.8951	0.3608
Jinjiang (JJ)	0.2727	0.6033	0.3967	1.6575	0.3180
Wannan (WN)	0.4167	0.5139	0.4861	1.9459	0.3680
Weining (GZWN)	0.3500	0.5450	0.4550	1.8349	0.3515
Zaobei (ZB)	0.2000	0.6800	0.3200	1.4706	0.2688
Dabeishan (DBS)	0.2167	0.6606	0.3394	1.5139	0.2818
Bashan (BS)	0.2500	0.6250	0.3750	1.6000	0.3047
Leiqiong (LQ)	0.4286	0.5102	0.4898	1.9600	0.3698
Wengshan (WS)	0.5500	0.5050	0.4950	1.9802	0.3725
Dianzhong (DZ)	0.6000	0.5200	0.4800	1.9231	0.3648
Guangfeng (GF)	0.1500	0.7450	0.2550	1.3423	0.2225
Sanjiang (SJ)	0.2500	0.6250	0.3750	1.6000	0.3047
Wuling (WL)	0.3308	0.5573	0.4427	1.7944	0.3447
Guanling (GL)	0.3500	0.5450	0.4550	1.8349	0.3515
Wuchuan (WC)	0.2586	0.6165	0.3835	1.6220	0.3099
Minnan (MN)	0.3000	0.5800	0.4200	1.7241	0.3318
Longling (LL)	0.3462	0.5473	0.4527	1.8270	0.3502
Weizhou (WZ)	0.5536	0.5057	0.4943	1.9773	0.3721
Nandan (ND)	0.3400	0.5512	0.4488	1.8142	0.3481
Diqing (DQ)	0.0833	0.8472	0.1528	1.1803	0.1411
Yunnan Humped (DH)	0.7368	0.6122	0.3878	1.6335	0.3126
Zhaotong (ZT)	0.2708	0.6050	0.3950	1.6528	0.3170
Yunling (YL)	0.4200	0.5128	0.4872	1.9501	0.3685
Tibetan (TB)	0.0000	1.0000	0.0000	1.0000	0.0000
Shigatse Humped (SH)	0.3833	0.5272	0.4728	1.8967	0.3610
Burma (MD)	0.7167	0.5939	0.4061	1.6838	0.3236
Angus (AG)	0.0000	1.0000	0.0000	1.0000	0.0000

**Table 2 animals-09-00915-t002:** Genotypic and allele frequencies of the *MTOR* genes across 37 cattle breeds.

Geographical Grouping	Breeds (Codes)	*MTOR* (NC_037343.1:c. 2062G>C)					Mutation Frequencies
		Genotype Frequencies (Number)			Allele Frequencies		
		GG	GC	CC	G	C	
Northern group	Kazakh (KZ)	0.08750 (21)	0.1250 (3)	0.0000 (0)	0.9375	0.0625	0.0625
	Yanbian (YB)	1.0000 (30)	0.0000 (0)	0.0000 (0)	1.0000	0.0000	0.0000
	Chaidamu (QD)	0.5333 (16)	0.4000 (12)	0.0667 (2)	0.7333	0.2667	0.2667
	Mongolian (MG)	0.7714 (27)	0.2286 (8)	0.0000 (0)	0.8857	0.1143	0.1143
		0.7899 (94)	0.1933 (23)	0.0168 (2)	0.8866	0.1134	0.1134
Central group	Jiaxian Red (JX)	0.5667 (17)	0.4000 (12)	0.0333 (1)	0.7667	0.2333	0.2333
	Nanyang (NY)	0.4737 (9)	0.4737 (9)	0.0526 (1)	0.7105	0.2895	0.2895
	Luxi (LX)	0.5172 (15)	0.4138 (12)	0.0690 (2)	0.7241	0.2759	0.2759
	Bohai Black (BH)	0.6000 (18)	0.2667 (8)	0.1333 (4)	0.7333	0.2667	0.2667
	Jinnan (JN)	0.5000 (6)	0.2500 (3)	0.2500 (3)	0.6250	0.3750	0.3750
	Qinchuan (QC)	0.7500 (21)	0.2143 (6)	0.0357 (1)	0.8571	0.1429	0.1429
		0.5811 (86)	0.3378 (50)	0.0811 (12)	0.7500	0.2500	0.2500
Southern group	Ji’an (JA)	0.3529 (6)	0.5294 (9)	0.1176 (2)	0.6176	0.3824	0.3824
	Jinjiang (JJ)	0.4545 (5)	0.5454 (6)	0.0000 (0)	0.7273	0.2727	0.2727
	Wannan (WN)	0.3333 (10)	0.5000 (15)	0.1667 (5)	0.5833	0.4167	0.4167
	Weining (GZWN)	0.3667 (11)	0.5667 (17)	0.0667 (2)	0.6500	0.3500	0.3500
	Zaobei (ZB)	0.6000 (6)	0.4000 (4)	0.0000 (0)	0.8000	0.2000	0.2000
	Dabeishan (DBS)	0.6333 (19)	0.3000 (9)	0.0667 (2)	0.7833	0.2167	0.2167
	Bashan (BS)	0.5227 (23)	0.4545 (20)	0.0227 (1)	0.7500	0.2500	0.2500
	Leiqiong (LQ)	0.3061 (15)	0.5306 (26)	0.1633 (8)	0.5714	0.4286	0.4286
	Wengshan (WS)	0.1000 (3)	0.7000 (21)	0.2000 (6)	0.4107	0.5893	0.5893
	Dianzhong (DZ)	0.1333 (4)	0.5333 (16)	0.3333 (10)	0.4000	0.6000	0.6000
	Guangfeng (GF)	0.7000 (7)	0.3000 (3)	0.0000 (0)	0.8500	0.1500	0.3000
	Sanjiang (SJ)	0.5385 (14)	0.4231 (11)	0.0385 (1)	0.7500	0.2500	0.2500
	Wuling (WL)	0.4923 (32)	0.3538 (23)	0.1538 (10)	0.6692	0.3308	0.3308
	Guanling (GL)	0.5000 (15)	0.3000 (9)	0.2000 (6)	0.6500	0.3500	0.3500
	Wuchuan (WC)	0.6207 (18)	0.2414 (7)	0.1379 (4)	0.7414	0.2586	0.2586
	Minnan (MN)	0.4000 (6)	0.6000 (9)	0.0000 (0)	0.7000	0.3000	0.3000
	Longling (LL)	0.3846 (5)	0.5385 (7)	0.0769 (1)	0.6538	0.3462	0.3462
	Weizhou (WZ)	0.2500 (7)	0.3929 (11)	0.3571 (10)	0.4464	0.5536	0.5536
	Nandan (ND)	0.4400 (11)	0.4400 (11)	0.1200 (3)	0.6600	0.3400	0.3400
	Diqing (DQ)	0.8667 (26)	0.1000 (3)	0.0333 (1)	0.9167	0.0833	0.0833
	Yunnan Humped (DH)	0.0000 (0)	0.5263 (10)	0.4737 (9)	0.2632	0.7368	0.7368
	Zhaotong (ZT)	0.5625 (27)	0.3333 (16)	0.1042 (5)	0.7292	0.2708	0.2708
	Yunling (YL)	0.2800 (7)	0.6000 (15)	0.1200 (3)	0.5800	0.4200	0.4200
		0.4283 (275)	0.4330 (278)	0.1386 (89)	0.6449	0.3551	0.3551
Special	Tibetan (TB)	1.0000 (29)	0.0000 (0)	0.0000 (0)	1.0000	0.0000	0.0000
	Shigatse Humped (SH)	0.40000 (12)	0.4333 (13)	0.1667 (5)	0.6167	0.3833	0.3833
		0.6949 (41)	0.2203 (13)	0.0848 (5)	0.8051	0.1949	0.1949
Exotic	Over all	0.5134 (498)	0.3753 (364)	0.1113 (108)	0.7010	0.2990	0.2990
	Burma (MD)	0.0667 (2)	0.4333 (13)	0.5000 (15)	0.2833	0.7167	0.7167
	Angus (AG)	1.0000 (30)	0.0000 (0)	0.0000 (0)	1.0000	0.0000	0.0000

**Table 3 animals-09-00915-t003:** Association of the *MTOR* gene variation with temperature (T), relative humidity (RH) and the temperature humidity index (THI) in Chinese cattle.

SNP	Genotype (n)	T (°C) (LSM ± SE)	RH (%) (LSM ± SE)	THI (LSM ± SE)
*MTOR* :NC_037343.1:c.2062G>C	CC(108)	16.06 ^A^ ± 0.54	74.43 ^A^ ± 0.99	60.19 ^A^ ± 0.76
	GC(364)	15.12 ^A^ ± 0.29	72.89 ^A^ ± 0.59	59.07 ^A^ ± 0.41
	GG(498)	12.05 ^B^ ± 0.25	68.53 ^B^ ± 0.49	54.60 ^B^ ± 0.35

LSM ± SE, least squares means and their standard errors for each genotypic class reported. Uppercase letters mean differences of the value at *p* < 0.01. SNP, the single nucleotide polymorphism; T, the mean annual temperature; RH, the relative humidity; THI, the temperature humidity index.

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
