# Peer review of "MTOR* Variation Related to Heat Resistance of Chinese Cattle"

_animals, 2019, doi:10.3390/ani9110915_

Round 1

Reviewer 1 Report

Because this is the second review , I can only say that authors did a great job, thus increasing  the value of the manuscript. At the same time, the authors comprehensively replied to the comments I sent. I am not sure exactly what the magazine's requirements are, but it would be good if the text was checked by a native speaker. I am not a language expert and I am not trying to correct the text, but I feel that it still needs to be improved. I have no other comments and I think the work qualifies for printing in Animals.

Reviewer 2 Report

The revised manuscript entitled “MTOR variation related to heat resistance of Chinese cattle” appears suitable for publication. I liked the response 5, and I think authors can consider response 5 as part of discussion. Overall, manuscript can be accepted.

This manuscript is a resubmission of an earlier submission. The following is a list of the peer review reports and author responses from that submission.

Round 1

Reviewer 1 Report

            Global temperature is increasing at an alarming rate. This is a serious threat to humanity and livestock. Animals continuously adapt to such environmental changes and it is a continuous process. Therefore, there is a global need to screen the environmental impact on humans and livestock. As described by Jirimutu et al., (2012) camels are adapted to high salt and temperature conditions. Here, Ning et al manuscript entitled “MTOR variation related to heat resistance of Chinese cattle” describes association between temperature and MTOR gene polymorphism. Authors have done a comprehensive job of screening 1030 animals from 37 cattle breed across China. Authors screened for ‘NC_037343.1:c.2062G>C’ locus in MTOR gene, and reported high association of C with higher temperature and a clear association between G and lower temperature (YB, TB, Angus). Overall, the study delineate the criteria of genetic selection for heat resistant cattle. However, few aspects need to be addressed or clarified to provide a clearer picture of this work.

Concerns:

Please maintain uniform nomenclature throughout the manuscript; ‘Zebu’ has been called with different names. The study uses Angus and Burmese Zebu as a control. However, the reason for such selection need to be mentioned in the text. Did author apply any formula to determine the sample size in each breed? Although MTOR locus (NC_037343.1:c.2062G>C) is mentioned few times, the justification or references for choosing this particular locus is never mentioned in the text. Also, what are the amino acid changes expected in G/C substitution? And how does it affect the mTOR activity? In the discussion section, line 173, authors talk about thermal diseases in animals, examples of such diseases are needed. The line “MTOR gene had been proved to repair DNA damage via regulation of the expression of DNA damage response enzymes” is repetitive. And give examples of enzymes regulated by mTOR. 179-181 describes findings from current manuscript, and says as reported by reference number 24, 25. Hansen et al and Zeng et al do not report MTOR Therefore, this paragraph need to be rewritten. Overall discussion section can be improved Please rewrite figure 1 legend. Figure and legend do not match, there is no a, b, c in the figure. Do circled pie charts represent allelic frequencies? Do different colors mean anything? For better understanding, authors need to correctly and clearly explain different parts of HWE formula in the context of present study.

Author Response

Point 1: Overall, the study delineates the criteria of genetic selection for heat resistant cattle. However, few aspects need to be addressed or clarified to provide a clearer picture of this work. Response 1: Thanks for your kind comments on my manuscript. This study is to aim to speculate a novel and screened gene–MTOR for marker-assisted selection related to the heat tolerance. Next, these doubtful and unclear points will be point-by-point response. Point 2: Please maintain uniform nomenclature throughout the manuscript; ‘Zebu’ has been called with different names. Response 2: I will use uniform name “Bos indicus” or “indicine cattle” so that my paper becomes more rigorous. Thanks for your suggestion. Point 3: The study uses Angus and Burmese Zebu as a control. However, the reason for such selection needs to be mentioned in the text. Response 3: Done as required. Please see line 94-96. Point 4: Did author apply any formula to determine the sample size in each breed? Response 4: There is no any formula when we select sample size in each breed, and it also believed to be random. There are two main reasons: firstly, sample size collection can’t be assured due to different geographic and cultural barriers, and it will happen that some sizes are large but others are small; Secondly, though this study is to speculate the association between individuals and the mutation of MTOR gene, the subjects of this study are according to the three large groups of the northern cattle (Bos taurus), central cattle (admixture of Bos taurus and Bos indicus)and the southern cattle (Bos indicius), emphasizing the heat-resistance distribution of Chinese yellow cattle. Point 5: Although MTOR locus (NC_037343.1:c.2062G>C) is mentioned few times, the justification or references for choosing this particular locus is never mentioned in the text. Also, what are the amino acid changes expected in G/C substitution? And how does it affect the MTOR activity? Response 5: Thanks for your valuable suggestion. In the revision, we have added several published papers to explain the gene function more clearly. This SNP is located in ~1.1kb downstream of MTOR gene (NC_037343.1:c.2062G>C, rs445599276). And it occurs in non-coding region. We reviewed others paper about the related functions of gene downstream regulation and found the downstream area also plays a very important role. For example, Verkuil et al (2018) demonstrated that a SNP downstream of TMCO1, rs4657473, was associated with primary open-angle glaucoma in an African Americans population. Mijac et al (2016) found that rs3024505 (C/T) located downstream of the IL10 gene was associated with both ulcerative colitis and Crohn's disease in Western populations. Beghelli et al (2011) indicated that an A/T SNP corresponding to a LINE2 sequence located ~1.1 kb downstream of the IL-6 gene was associated with the basal Cu:Zn ratio in horses. Therefore, this SNP is located in non-coding region, but we thought it could be still helpful for heat tolerance. Verkuil, Lana,; Danford, Ian.;Pistilli, Maxwell.; Collins, David W..; Gudiseva, Harini V.; . . .;O'Brien, Joan Marie. SNP located in an AluJb repeat downstream of TMCO1, rs4657473, is protective for POAG in African Americans, The British journal of ophthalmology, 2019, 103(10): 1530-1536. Mijac D , Petrovic I V.; Djuranovic S , et al. The Polymorphism rs3024505 (C/T) Downstream of the IL10 Gene Is Associated with Crohn’s Disease in Serbian Patients with Inflammatory Bowel Disease. The Tohoku Journal of Experimental Medicine, 2016, 240(1):15-24. Beghelli D.;, Giacconi R.; Mocchegiani E , et al. A genetic variant near the equine interleukin 6 gene associated with copper:zinc ratio. Veterinary Journal, 2011, 190(2): e143-5. Point 6: In the discussion section, line 173, authors talk about thermal diseases in animals, examples of such diseases are needed. The line “MTOR gene had been proved to repair DNA damage via regulation of the expression of DNA damage response enzymes” is repetitive. And give examples of enzymes regulated by MTOR. Therefore, this paragraph needs to be rewritten. Response 6: Thanks for your valuable comments. I have rewritten this paragraph as suggested. Please see line 194 (the discussion section). Point 7: 179-181 describes findings from current manuscript, and says as reported by reference number 24, 25. Hansen et al and Zeng et al do not report MTOR. Response 7: Thanks for your detailed suggestion. I have checked out and modified this. Point 8: Please rewrite figure 1 legend. Figure and legend do not match, there is no a, b, c in the figure. Do circled pie charts represent allelic frequencies? Do different colors mean anything? Response 8: Sorry, I wish to explain figure 1 legend and means of “a, b, c” before rewriting. If you still feel these descriptions are not clear, I will rewrite and rearrange. “a” is to explain the composition of the total sample size in this map, also means that this study includes 35 Chinese breeds as well as Angus and Burma population (two controls). “b” is to explain distribution of G allele and C allele in every breed and detailed information of the MTOR gene as well as so that we can clearly know geographic distribution of allelic information. “c” is to explain the full name of each breeds abbreviation on the map so that our sample display is more precise. Circled pie charts represent allelic frequencies, and different colors mean different alleles. As the lower left corner of the figure are showed, it explains that the blank padding is G allele but the black padding is C allele in the pie charts. Point 9: For better understanding, authors need to correctly and clearly explain different parts of HWE formula in the context of present study. Response 9: Thanks for your question. I have done as suggested in the revision.

Reviewer 2 Report

Despite the theme of the present manuscript is within the scope of the journal and the work deals a potential interesting topic, the description of the investigation and data analysis are absolutely inadequate.

In my opinion, this research needs a complete reconceptualization and reorganization.

The present manuscript has a lot of negative aspects that preclude its acceptance in a scientific journal.

Author Response

Response: Thanks for your comments. This MTOR gene is never reported about heat tolerance and this SNP of MTOR is novel and screened. Based on previous studies, our research has been speculated to be feasible and the result is also optimistic. So this research can provide a novel and helpful pathway for marker-assisted selection related to the heat tolerance. We sincerely hope that you can give our paper a chance to show its weak value.

Reviewer 3 Report

The paper  sent to me for evaluation presents a very important problem of heat stress and its impact on livestock agriculture. The study was conducted on a large group of animals and comprehensively covered a large area of China.

In my opinion, the paper does not present a high scientific value. Although the authors dealt with the very current topic of heat stress and its impact on animals, they limited themselves to examining only one gene  and, what more important, only one polymorphism within this gene. As I understand it, this mutation has been described previously and associated with thermal stress, so the authors did not bring much new information, they only confirmed that allele “C” showed a higher frequency in the area  with higher average annual temperatures (in this case southern-western population). In an era when molecular techniques allow us to study the precise mechanisms of gene activity, limiting ourselves to exploring a single polymorphism is the shortcut. What about other mutations within this gene ? what about the expression of this gene, other regulations ??? The authors do not write anything closer about the polymorphism itself, its location (exon, intron?) and the possible effects of this mutation on protein production: is it a nonsense type change or something else???

As for the methodology, I also do not understand why all the samples were sequenced. It would be much simpler and cheaper to perform the ordinary RFLP method. Of course, you would have to find the enzyme cutting in or next to SNP, but still, this is the simplest method. Especially when we have PCR product of the weight only 571 bp.( medium size).

In realtion to the number of samples. In one place (line 79) authors wrote about 1030 cows/samples while in another place (line 161) only 970. Where did the difference come from? This should be clarified.

English needs improvement. A lot of articles are missing, sample text below (the words in bold were added or replace):

In recent years, the frequency of thermal diseases in animals has risen with the increase of the global 173 temperature. The Selection of heat-resistant traits is necessary in the commercial cattle industry. Previous 174 study results showed that the rapidly evolving MTOR gene in for camels plays an important role in 175 adapting the dryness-heat environment [12]. Also, the MTOR gene had been proved to repair DNA 176 damage via regulation of the expression of DNA damage response enzymes [10,11]. Therefore, the 177 MTOR is an important candidate gene responsible for heat tolerance in animals. 178

Furthermore, the distribution of allelic frequencies in control groups showed that the allele G of 179 the MTOR gene is dominant in B.taurus cattle (Angus), whereas the C allele is dominant in B. indicus 180 cattle (Burmese zebu) [24,25]. The A similar pattern was observed in the Chinese native cattle population. 181 Results also showed that the frequencies of G allele of the MTOR gene diminished gradually in native 182 Chinese cattle from the northern group to southern group, whereas the frequencies of the C allele 183 showed an opposite pattern (Fig.1), indicating a significant geographical difference across native 184 Chinese cattle breeds and consistent with the distribution of indicine and taurine cattle in China. For 185 the southern group, the highest frequencies of the C allele of the MTOR gene were found in cattle in 186 southwestern China, which has the highest temperatures compared to other regions, followed by 187 cattle is southeastern China. Correspondingly, we also found that cattle breeds with particularly high 188 C allele frequency had higher mean annual temperatures in the southern region. Our results were 189 consistent with previous studies that revealed the geographical environment and origin of Chinese 190 indigenous cattle [24-27].

And a few others:

-line 28 shown, except showed

-line 34 the southern group

line  48 the earths

-  In line121 – locus except loci

-In line 166 - For me the sentence …”effect of the three studied parameters on MTOR genotype showed that mean annual temperature had the strongest correlation with the genotypes” is illogical. It will be more logical if genotype will be depending of temperature.

 -etc.

Author Response

Point 1: In my opinion, the paper does not present a high scientific value. Although the authors dealt with the very current topic of heat stress and its impact on animals, they limited themselves to examining only one gene and, what more important, only one polymorphism within this gene.

Response 1: Maybe one polymorphic function is too tiny to explain related problem. But this SNP of the MTOR gene is novel and screened, and its value provides a more selection for marker-assisted selection related to the heat tolerance.

Point 2: As I understand it, this mutation has been described previously and associated with thermal stress, so the authors did not bring much new information, they only confirmed that allele “C” showed a higher frequency in the area  with higher average annual temperatures (in this case southern-western population).

Response 2: This mutation has not been described previously relating to thermal stress. This article aims to explain the association of allele “C” and individuals with heat-resistant phenotypic characteristics which are reflected by corresponding to the living environmental temperature and humidity. Such as, our results exhibited that allele “C” existed a highest frequency (DH: 0.7368; WS: 0.5893) in the southwestern China, whereas allele “G” showed a highest frequency (YB: 1.00) in the northeastern China.

Point 3: In an era when molecular techniques allow us to study the precise mechanisms of gene activity, limiting ourselves to exploring a single polymorphism is the shortcut. What about other mutations within this gene?

Response 3: Other mutations within this gene have nearly similarly significant geographical distribution rule in Chinese cattle from the BGVD (Bovine Genome Variation Database and Selective Signatures), but only the SNP of this study owned the supports for related literature.

Point 4:  The authors do not write anything closer about the polymorphism itself, its location (exon, intron?) and the possible effects of this mutation on protein production: is it a nonsense type change or something else???

Response 4: This SNP is located in ~1.1kb downstream of MTOR gene (NC_037343.1:c.2062G>C, rs445599276). We reviewed other papers about the related functions of gene downstream regulation and found the downstream region also plays a very important role. For example, Verkuil et al (2018) demonstrated that a SNP downstream of TMCO1, rs4657473, is associated with primary open-angle glaucoma in an African Americans population. Mijac et al (2016) found that rs3024505 (C/T) located downstream of the IL10 gene was associated with both ulcerative colitis and Crohn's disease in Western populations. Beghelli et al (2011) indicated that an A/T SNP corresponding to a LINE2 sequence located ~1.1 kb downstream of the IL-6 gene was associated with the basal Cu:Zn ratio in horses. Therefore, this SNP is located in non-coding region, but we thought it could be still helpful for heat tolerance.

Verkuil, Lana,; Danford, Ian.;Pistilli, Maxwell.; Collins, David W..; Gudiseva,    Harini V.; . . .;O'Brien, Joan Marie. SNP located in an AluJb repeat downstream of TMCO1, rs4657473, is protective for POAG in African Americans, The British journal of ophthalmology, 2019, 103(10): 1530-1536.

Mijac D , Petrovic I V.; Djuranovic S , et al. The Polymorphism rs3024505 (C/T) Downstream of the IL10 Gene Is Associated with Crohn’s Disease in Serbian Patients with Inflammatory Bowel Disease. The Tohoku Journal of Experimental Medicine, 2016, 240(1):15-24.

Beghelli D.;, Giacconi R.; Mocchegiani E , et al. A genetic variant near the equine interleukin 6 gene associated with copper:zinc ratio. Veterinary Journal, 2011, 190(2): e143-5.

Point 5: As for the methodology, I also do not understand why all the samples were sequenced. It would be much simpler and cheaper to perform the ordinary RFLP method. Of course, you would have to find the enzyme cutting in or next to SNP, but still, this is the simplest method. Especially, when we have PCR products of the weight only 571 bp (medium size).

Response 5: The samples were sequenced because the results of sequencing method are most accurate and the budget of this study is within our fund plan. The RFLP operation is complicated, impossible to carry out multiple reactions, and has low flux, which is only suitable for the SNP where the enzyme cleavage site exists. However, there isn’t an enzyme cutting site in this study, more importantly SNP is easy to implement automated analysis so that it only needs to determine "with or without" when analyzing and testing.

Point 6: In relation to the number of samples, in one place (line 79) authors wrote about 1030 cows/samples while in another place (line 161) only 970. Where did the difference come from? This should be clarified.

Response 6: The first place (line 79) represents total number of samples (1030 individuals) which includes 35 Chinese native cattle breeds as well as Angus and Burmese cattle as controls (there are 37 breeds in all); The second place (line 161) shows 970 Chinese indigenous individuals (except the number of samples as controls) which have corresponding value of T, H, THI.

Point 7: English needs improvement.

Response 7: In the revised version, I have made changes to the inappropriate expression. Thank you for your suggestion.

Point 8: A lot of articles are missing, sample text below (the words in bold were replaced).

Response 8: Thanks for your kind suggestions. I have done as required in the review. Please see the places in red.
